# Generalisable deep learning method for mammographic density prediction across imaging techniques and self-reported race

Galvin Khara [1] ✉, Hari Trivedi[2], Mary S. Newell[2], Ravi Patel[1], Tobias Rijken [1], Peter Kecskemethy[1] & Ben Glocker [1,3] ✉

## Abstract

**Background** Breast density is an important risk factor for breast cancer complemented by a higher risk of cancers being missed during screening of dense breasts due to reduced sensitivity of mammography. Automated, deep learning-based prediction of breast density could provide subject-specific risk assessment and flag difficult cases during screening. However, there is a lack of evidence for generalisability across imaging techniques and, importantly, across race.

**Methods** This study used a large, racially diverse dataset with 69,697 mammographic studies comprising 451,642 individual images from 23,057 female participants. A deep learning model was developed for four-class BI-RADS density prediction. A comprehensive performance evaluation assessed the generalisability across two imaging techniques, full-field digital mammography (FFDM) and two-dimensional synthetic (2DS) mammography. A detailed subgroup performance and bias analysis assessed the generalisability across participants' race.

**Results** Here we show that a model trained on FFDM-only achieves a 4-class BI-RADS classification accuracy of 80.5% (79.7–81.4) on FFDM and 79.4% (78.5–80.2) on unseen 2DS data. When trained on both FFDM and 2DS images, the performance increases to 82.3% (81.4–83.0) and 82.3% (81.3–83.1). Racial subgroup analysis shows unbiased performance across Black, White, and Asian participants, despite a separate analysis confirming that race can be predicted from the images with a high accuracy of 86.7% (86.0–87.4).

**Conclusions** Deep learning-based breast density prediction generalises across imaging techniques and race. No substantial disparities are found for any subgroup, including races that were never seen during model development, suggesting that density predictions are unbiased.

## Plain language summary

Women with dense breasts have a higher risk of breast cancer. For dense breasts, it is also more difficult to spot cancer in mammograms, which are the X-ray images commonly used for breast cancer screening. Thus, knowing about an individual's breast density provides important information to doctors and screening participants. This study investigated whether an artificial intelligence algorithm (AI) can be used to accurately determine the breast density by analysing mammograms. The study tested whether such an algorithm performs equally well across different imaging devices, and importantly, across individuals from different self-reported race groups. A large, racially diverse dataset was used to evaluate the algorithm's performance. The results show that there were no substantial differences in the accuracy for any of the groups, providing important assurances that AI can be used safely and ethically for automated prediction of breast density.

Mammographic density is one of the strongest risk factors for breast cancer, with an almost five-fold increase in risk for women in the highest breast density group compared to women of similar age in the lowest group[1].

Breast density also affects the accuracy of breast cancer detection in screening mammography, as normal but dense parenchymal breast tissue can have similar radiographic appearance as cancerous tissue[2,3]. The

[1]Kheiron Medical Technologies, London, UK. [2]Winship Cancer Institute, Emory University, Atlanta, GA, USA. [3]Department of Computing, Imperial College London, London, UK. ✉e-mail: galvin@kheironmed.com; b.glocker@imperial.ac.uk

reduced sensitivity of mammography to detect cancer in highly dense breasts makes mammographic density an important risk factor for interval cancers[4]. Automating the assessment of breast density from mammographic images would not only provide important information about subject-specific cancer risk, but could be used to flag difficult cases during screening which may require additional attention. Breast density in mammographic images is commonly assessed following The Breast Imaging Reporting and Data System (BI-RADS) standard[5], which defines four levels of density: A, almost entirely fatty; B, scattered areas of fibroglandular density; C, heterogeneously dense; and the highest level D, extremely dense. Despite this widely used reporting standard, human readers exhibit high inter-rater variability in breast density classification, questioning its utility for clinical considerations and health policy[6–8]. Given its clinical relevance and the recent US federal mandate requiring all screening mammograms to include a density assessment[9], improving its reliability and reproducibility could contribute to a decrease in breast cancer mortality[10].

Computational methods for image-based density assessment could provide more objective and reliable measurements, and thus, improve its value in clinical decision making. Early approaches for interactive density quantification showed potential, enabling research into better understanding the association of breast density and cancer risk[11]. One study found excellent reproducibility of automated methods[12], arguing that computer algorithms could be well suited for inclusion in breast cancer risk prediction models[13].

More recently, deep learning-based methods have been proposed for full four-class BI-RADS classification[14–18], and binary dense/non-dense classification using full-field digital mammography (FFDM)[19,20], with promising performance and good agreement with expert radiologists. In clinical practice, both FFDM and two-dimensional synthetic (2DS) images generated from digital breast tomosynthesis (DBT) are used in screening and diagnostic imaging. Previous research highlighted human perceptible differences, and higher signal-to-noise ratio, for 2DS images, when viewing parenchymal tissue, a key contributor to the BI-RADS classification[21]. Automated systems for density classification should be accurate and reliable, independent of the underlying imaging technique. A recent multi-site study demonstrated that a deep learning system trained on FFDM could achieve a good level of accuracy when tested on 2DS images[17]. However, a model adaptation step was required where the FFDM model had to be fine-tuned using 2DS training data. This may limit its clinical adoption as different models would need to be deployed depending on the imaging technique used at a local site.

To be clinically effective and ethical to use in practice, automated breast density prediction should not only be robust to changes in the image acquisition, but more importantly, it should generalise across different patient populations. Differences in breast density across race and its association with cancer risk has been the subject of many research studies[22–26]. Despite its clinical and societal relevance, performance across race has not been studied in the context of automated breast density prediction. None of the recent works on deep learning-based methods have evaluated the potential effect of race nor provided any assurances that the proposed methods are unbiased and yield equitable performance across subgroups. This is concerning in the light of recent work that has demonstrated that AI systems can predict race from mammograms[27]. The ability of deep learning models to recognise such protected attributes exposes the models to potentially harmful shortcuts that could be exploited during model training[28]. Existing biases in the training data related to race could be picked up and encoded in the model, possibly leading to disparate performance across subgroups. The presented work makes an important contribution by validating the generalisability of a deep learning-based breast density prediction method across imaging techniques and race.

This study used a large, racially diverse dataset to assess the performance of a deep learning model for four-class BI-RADS density prediction. A comprehensive performance evaluation assessed the generalisability across FFDM and 2DS images. A detailed subgroup performance analysis assessed the generalisability across participants' self-reported race.

The racial subgroup analysis shows unbiased performance across Black, White, and Asian participants. The study results suggest that deep learning-based breast density prediction generalises across imaging techniques and race, including races that were never seen during model development. The study provides important assurances that AI can be used safely and ethically for automated prediction of breast density from mammographic images.

## Methods

### Study population

We used a large scale representative dataset with 69,697 mammographic imaging studies with a total of 451,642 individual images from a cohort of 23,057 female participants, collected from four hospital sites at Emory University (Atlanta, GA USA) between 2013 and 2020 as part of the EMory BrEast imaging Dataset (EMBED)[29]. The retrospective data was collected with the approval of the Emory University's institutional review board. The need for written informed consent from patients was waived because of the use of de-identified data. The EMBED dataset is made publicly available for research. The presented study is exempt from ethical approval as the analysis is based on this publicly available, fully anonymized data. All imaging studies were acquired on a variety of Hologic Selenia and Selenia Dimensions machines, consisting of standard FFDM only systems, as well as newer systems with combo imaging capability, which contain both FFDM and 2DS images. Only images tagged with laterality left or right, and craniocaudal and mediolateral oblique views were included. All studies required a historical BI-RADS density assessment (5th edition) by a human expert, and a corresponding self-reported race. Images flagged as having implants, or having a prior history of breast cancer, were not included. A valid study was composed of any images that satisfy all the above criteria. The historical BI-RADS assessments were made by 57 historical Mammography Quality Standards Act and Program (MQSA) certified, fellowship-trained breast radiologists, with 13 of these having read over 90% of the studies. A detailed breakdown of the population characteristics is provided in Table 1.

### Race subgroups

Self-reported race was available for all participants. This study focused on three racial groups with participants who identified as either 'White or Caucasian' (shortened to 'White' in the following), 'Black or African American' (shortened to 'Black' in the following) and 'Asian'. Participants were randomly assigned to training, validation, and test sets in approximately a 50/10/40% ratio. Training and validation sets were used for model development, and the test set was solely used for the final model performance evaluation. All participants with race reported as Asian were assigned to the test set, maximising the sample size and reliability of performance metrics on this underrepresented subgroup. This also facilitated the assessment of the generalisability of the density prediction method on a race never seen during model development. A breakdown of the characteristics of the training, validation and test sets is provided in Table 2.

### Density prediction models

To study the generalisability of breast density prediction, we trained several models on different subsets of the training and validation data. All models were based on the ResNet-34 architecture, a widely used deep learning model for image classification[30]. The final output layer was adapted for the prediction of four BI-RADS density classes. To obtain a density prediction per imaging study, averaging was applied to the class probabilities obtained for the individual images.

Figures 1 and 2 highlight the substantial qualitative and quantitative differences between FFDM and 2DS images respectively. To assess the generalisability across these imaging techniques, we compared two density prediction models, one trained on FFDM studies only and one trained on all available studies, which included combination studies, containing both FFDM and 2DS images. The FFDM-only model was developed with training and validation sets consisting of 19,323 and 2929 studies, respectively. The training and validation sets for the second model included an additional 16,770 and 3053 combination studies for a total of 36,093 and

**Table 1 | Characteristics of the study population**

| Variable | | All | Black | White | Asian |
|---|---|---|---|---|---|
| Cases | Participants | 23,057 | 11,663 | 9824 | 1570 |
| | Studies | 69,697 | 36,501 | 29,612 | 3584 |
| | Images | 451,642 | 242,911 | 185,368 | 23,363 |
| Age (years) | <40 | 1967 (3) | 1071 (3) | 728 (2) | 168 (5) |
| | 40–49 | 15,386 (22) | 8307 (23) | 5687 (20) | 1392 (39) |
| | 50–59 | 18,756 (27) | 10,256 (28) | 7472 (25) | 1028 (29) |
| | 60–69 | 19,436 (28) | 10,270 (28) | 8470 (29) | 696 (19) |
| | 70–79 | 11,502 (16) | 5436 (15) | 5790 (20) | 276 (8) |
| | 80–89 | 2496 (4) | 1084 (3) | 1388 (5) | 24 (1) |
| | 90+ | 154 (<1) | 77 (<1) | 77 (<1) | 0 (0) |
| Imaging technique | FFDM | 35,587 | 18,158 | 15,825 | 1604 |
| | FFDM + 2DS | 34,110 | 18,343 | 13,787 | 1980 |
| BI-RADS density | A | 7945 (11) | 5261 (14) | 2599 (9) | 85 (2) |
| | B | 29,102 (42) | 16,627 (46) | 11,599 (39) | 876 (25) |
| | C | 28,725 (41) | 13,087 (36) | 13,543 (46) | 2095 (58) |
| | D | 3925 (6) | 1526 (4) | 1871 (6) | 528 (15) |

Breakdown for BI-RADS density and imaging technique is given as the number of studies. Percentages are given in brackets. All numbers above are study counts, unless explicitly stated.
*FFDM* full-field digital mammography, *2DS* two-dimensional synthetic.

**Table 2 | Characteristics of the training, validation and test sets**

| Variable | | Training | Validation | Test |
|---|---|---|---|---|
| Cases | Participants | 11,832 | 1912 | 9313 |
| | Studies | 36,093 | 5982 | 27,622 |
| | Images | 230,954 | 39,494 | 181,194 |
| BI-RADS density | A | 4132 (11) | 720 (12) | 3093 (11) |
| | B | 15,433 (43) | 2660 (44) | 11,009 (40) |
| | C | 14,749 (41) | 2266 (38) | 11,710 (42) |
| | D | 1779 (5) | 336 (6) | 1810 (7) |
| Imaging technique | FFDM | 19,323 (53) | 2929 (49) | 13,335 (48) |
| | FFDM + 2DS | 16,770 (47) | 3053 (51) | 14,287 (52) |
| Race | Black | 19,152 (53) | 3404 (57) | 13,945 (50) |
| | White | 16,941 (47) | 2578 (43) | 10,093 (37) |
| | Asian | 0 (0) | 0 (0) | 3584 (13) |

Breakdown for BI-RADS density, imaging technique, and races is given as the number of studies. Percentages are given in brackets.
*FFDM* full-field digital mammography, *2DS* two-dimensional synthetic.

5982 studies, respectively. Both models were then evaluated on the same FFDM-only and 2DS-only test sets with 13,335 and 14,287 studies, respectively. This allowed us to assess two different aspects of generalisation. First, we evaluated the performance of the FFDM-only model on 2DS imaging in comparison to its performance on an FFDM test set. Second, we evaluated the added benefit of training on both FFDM and 2DS images in comparison to a model trained on FFDM-only.

To investigate the generalisation across races, we compared three breast density prediction models; model A trained on studies from Black patients only; model B trained on studies from White patients only; and model C trained on studies from both Black and White patients. Here, all three models were trained on both FFDM and 2DS images, and model C is identical to the FFDM/2DS model described above. The three models were then evaluated on the same large scale, diverse test set including participants that self-identified as either Black, White or Asian. This experimental setup allowed us to assess the robustness of the density prediction across race, and the generalisability of models when tested on patient subgroups that were not seen during development. Details about the BI-RADS proportions across subgroups in the test set are provided in Table 3.

**Bias analysis and model inspection**

To further investigate potential biases in the density prediction models, we conducted a comprehensive analysis following a recently proposed framework for model inspection[31]. The objective was to better understand whether race information may be used for making predictions for mammographic breast density. Firstly, we first trained a model for predicting race to confirm whether the mammographic images encode this protected characteristic, as reported in a recent study[27]. We used a ResNet-34 architecture with a final layer adjusted for binary classification. Here, we focused on the two classes of images from Black and White participants, which puts our results into context of previous work[25,27].

In addition, we investigated whether the density prediction models retained race information in the learned feature representations. For this purpose, we followed the standard transfer learning approach of freezing the density model backbone that generates the imaging features and replace the final prediction layer with a new prediction layer that is fine-tuned for the task of race classification[27]. Comparing the predictive performance on race when using density model backbones trained on different population subgroups allowed us to identify potential biases. In particular, we compared the race classification performance when using density backbones trained on single races (models A and B from above) versus a backbone trained on data from multiple races (model C). If the density model trained on multiple races had picked up race information during training, we may expect to see higher performance for a fine-tuned race prediction layer compared to the density models trained on a single race only. If we find no substantial differences between the performance on race classification when using density backbones that have and have not been exposed to race variations, we may conclude that race information has not been picked up during training[31].

To further assess whether any biases related to race encoded in the feature representations were learned by a density prediction model, we performed a visual model inspection. We utilised t-distributed stochastic neighbour embedding (t-SNE)[32], an algorithm for visualising high-dimensional data, to analyse the feature space of different density models. We processed all images in the test sets with the different density prediction models and retained the feature vectors produced for each image by the global average pooling layer (the last layer before the prediction layer). Using t-SNE for dimensionality reduction, we then visualised the feature embeddings for individual images processed by single and multi-race density models to investigate whether any distinct race clusters emerge. Finally, we constructed a t-SNE visualisation for the model trained specifically for race prediction. Overlaying BI-RADS and race information onto the various two-dimensional scatter plots, allowed us to assess whether any obvious associations between the two types of information had been encoded in the feature representations.

**Statistics and reproducibility**

Model performance was reported as class-wise classification accuracy and area under receiver operating characteristic curve (AUROC). The primary metric of clinical relevance for density prediction is accuracy. AUROC was reported where we directly compare to performance reported in the literature, both for density prediction and race classification. All performance metrics were calculated using bootstrapping with 1000 samples and 95% confidence intervals (CI) based on a continuous linear percentile approach[33]. Each bootstrapped sample consisted of 5000 studies (except

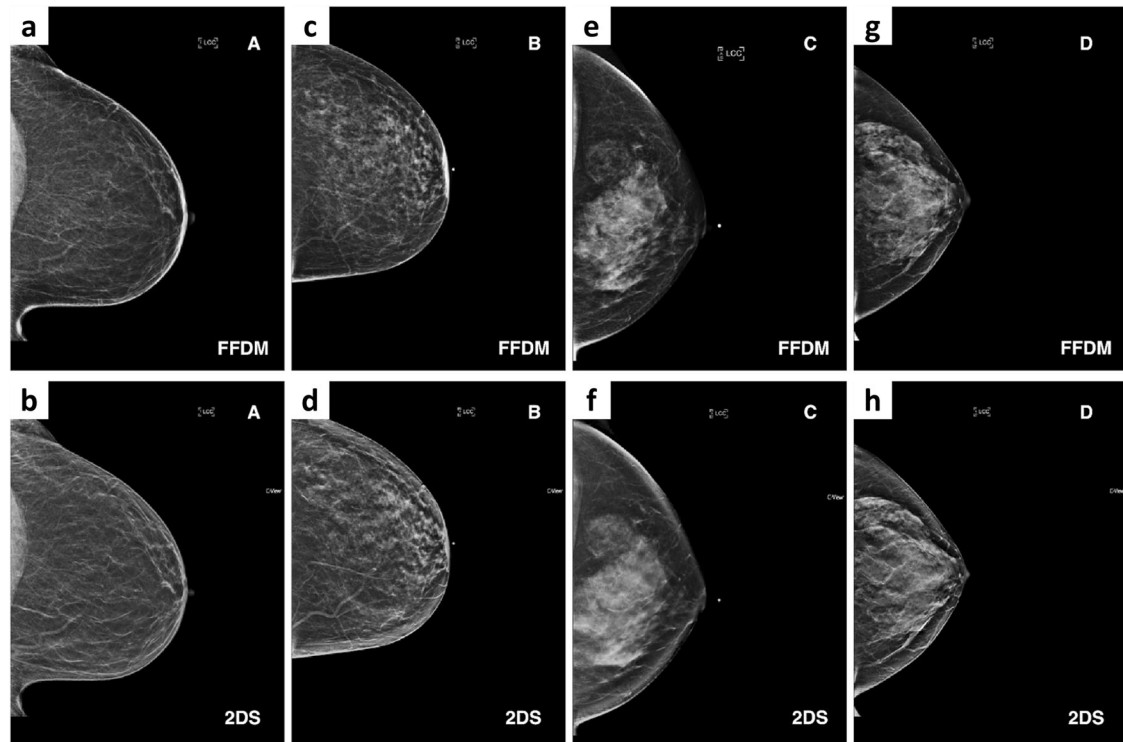

**Fig. 1 | Qualitative differences between FFDM and 2DS images.** Examples of the differences in image characteristics for FFDM in (**a, c, e, g**) and 2DS in (**b, d, f, h**) for image pairs from four participants. Columns show examples for the different BI-RADS density classes from A, almost entirely fatty in (**a, b**); B, scattered areas of fibroglandular density in (**c, d**); C, heterogeneously dense in (**e, f**); and the highest level D, extremely dense in (**g, h**). FFDM full-field digital mammography, 2DS two-dimensional synthetic.

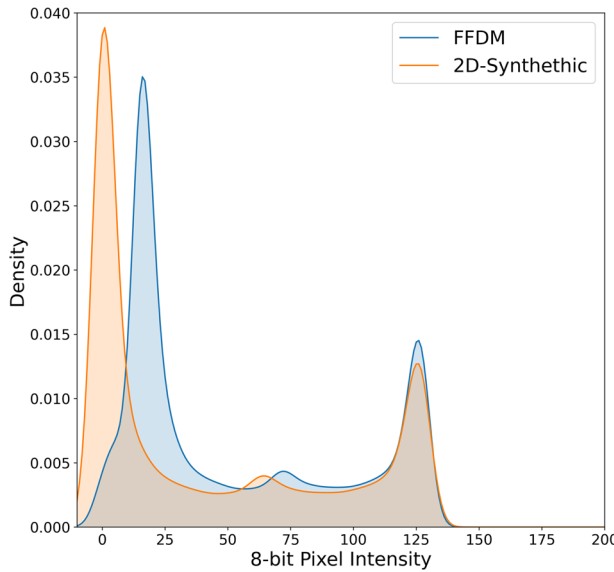

**Fig. 2 | Pixel intensity distributions for FFDM and 2DS images.** Average pixel intensity distributions on an 8-bit intensity range for FFDM and 2DS images on the same subset of combination studies. 2DS images are generally skewed towards the highest and lowest intensity values. FFDM full-field digital mammography, 2DS two-dimensional synthetic.

**Table 3 | BI-RADS density proportions across races in the test set**

| Variable | | Black | White | Asian |
|---|---|---|---|---|
| BI-RADS density | A | 2108 (15) | 900 (9) | 85 (2) |
| | B | 6177 (44) | 3956 (39) | 876 (25) |
| | C | 5059 (37) | 4556 (45) | 2095 (58) |
| | D | 601 (4) | 681 (7) | 528 (15) |

Breakdown for BI-RADS density is given as the number of studies. Percentages are given in brackets.

hypothesis assumes that the difference in the mean accuracy between populations is larger than the 2% bound. A *p*-value less than 0.05 (at a confidence level of 95%) suggests rejecting the null hypothesis, implying that the performance across both populations lies within this margin of equivalence. The single *p*-value reported was the largest value from either side of the TOST. All equivalence *p*-values were calculated from the bootstrapped samples. The implementation details provided next aim to facilitate reproducibility of the presented work.

**Implementation details**

All deep learning models in this work were based on the ResNet-34 architecture[30], implemented in TensorFlow 2. The same architecture was used in a recent study demonstrating state-of-the-art performance for breast density prediction[17]. The input to a model is a single mammographic image downsampled to 512 × 384 pixel resolution. Each model was trained for 200k iterations, with model checkpointing every 1000 iterations, using a batch size of 32 images. Training was performed on a single NVIDIA T100 GPU with 16GB RAM. We used the Adam optimizer with default parameters and a learning rate of 0.001[30,34]. The final inference model used a linear stochastic weight average of the last 25% of training iterations[35].

when evaluating on the Asian subgroup, which due to fewer overall participants, consisted of 500 studies), sampled with replacement. A 'two-one-sided t-tests' (TOST) procedure was used to test equivalence of performance, with an absolute equivalence margin of 2% for the difference in the average 4-class accuracy of compared groups. In this setup, the null

This has the advantages of ensembling over multiple models, without any increase in model complexity, training or inference time. When adapting the density models to perform race prediction, we froze all trainable parameters in the ResNet-34 up to and including the global average pooling layer, re-initialised the final dense layer to perform binary classification, and fine-tuned using the Adam optimizer, with default parameters, and a learning rate of 0.01. To generate the t-SNE plots, we used the openTSNE library with default parameters[36].

### Reporting summary
Further information on research design is available in the Nature Portfolio Reporting Summary linked to this article.

## Results
### Density prediction performance
Our primary model for mammographic density prediction that was developed on comprehensive training and validation sets, included both FFDM and 2DS images, and studies from Black and White participants, achieved a state-of-the-art 4-class accuracy for predicting BI-RADS density of 82.4% (95% CI 81.5–83.4) with a macro-AUROC of 96.0% (95.7–96.3). This is comparable or slightly better than previously reported performance of deep learning-based density prediction models (Table 4)[14–16,18].

**Table 4 | Density prediction performance in comparison to previous work**

| Model | 4-Class Accuracy (95% CI) | AUROC (95% CI) |
|---|---|---|
| Wu et al.[14] | 76.7% (??.?–??.?) | 91.6% (??.?–?.??) |
| Lehman et al.[16] | 77.?% (76.?–78.?) | ??.?% (??.?–?.??) |
| Matthews et al.[17] | 82.2% (81.6–82.9) | 95.2% (95.0–95.4) |
| Magni et al.[18] | 78.2% (??.?–??.?) | ??.?% (??.?–?.??) |
| Our model (FFDM/2DS) | 82.4% (81.5–83.4) | 96.0% (95.7–96.3) |

Performance comparison is indicative only as each evaluation was done on different test sets. The '?' indicates unknown values that were not reported in the literature.
*FFDM* full-field digital mammography, *2DS* two-dimensional synthetic.

**Table 5 | Generalisation of density prediction performance across imaging techniques**

| Training set | 4-Class Accuracy (95% CI) | |
|---|---|---|
| | FFDM test set | 2DS test set |
| FFDM-only | 80.5% (79.7–81.4) | 79.4% (78.5–80.2) |
| FFDM/2DS | 82.3% (81.4–83.0) | 82.3% (81.3–83.1) |

First row shows performance for a model trained on FFDM studies only. Second row is a model trained on FFDM and 2DS.
*FFDM* full-field digital mammography, *2DS* two-dimensional synthetic.

The model trained on FFDM studies only achieved a 1.8% lower accuracy compared to the model trained on FFDM and 2DS data. The training set for FFDM-only was much smaller, constituting 53% of all training data. The FFDM-only showed no substantial differences in accuracy when comparing performance on different test sets. The 4-class accuracy on the FFDM test set was 80.5% (79.7–81.4), and for the unseen 2DS test set it was 79.4% (78.5–80.2), demonstrating good generalisability across imaging techniques. Previous work reported drops in accuracy of 3.2–6.2% when training on FFDM and testing on 2DS[17]. The model trained on both FFDM and 2DS images showed an improved performance on both test sets, with an accuracy of 82.3% (81.4–83.0) on FFDM images, and 82.3% (81.3–83.1) on 2DS, demonstrating that adding 2DS data to the training was also beneficial for the FFDM performance. The density prediction results across imaging techniques are summarised in Table 5.

Comparing the density prediction across race, we found that the performance was comparable between all subgroups, while also consistent across the three models trained on different populations. We observed 4-class accuracies ranging between 80.2% (78.7–81.6) and 83.3% (80.6–86.8). Slight variations in the overall performance between models was observed in line with the changes in training set size. Models A (trained on Black-only) and B (trained on White-only) were each trained on only 53% and 47%, respectively, of the training data used for model C (which is our primary density prediction trained on data from Black and White participants). All three models generalised consistently to studies from the Asian subgroup, which was completely unseen during model development. The 4-class accuracies for Asian women were comparable to the performance on Black and White subgroups across all three density prediction models, despite the substantially different distributions across BI-RADS density classes (cf. Table 3). The density prediction results across race groups are summarised in Table 6. The results from the statistical tests for performance equivalence are reported in the Supplementary Tables 1–6, confirming the equitable performance across models and racial subgroups.

### Bias analysis results
The model trained specifically for race prediction achieved a classification accuracy of 86.7% (86.0–87.4) with an AUROC of 95.1% (94.7–95.5), confirming that race information is strongly encoded in the mammographic images. The race prediction models that were built upon the backbones of the density models A, B, and C showed substantially lower classification performance with each showing a drop in accuracy of about 10% and a drop in AUROC of about 15%. When comparing the absolute performance across the three race models built from fine-tuning density models, we observed comparable performance with no substantial differences across all three, indicating that the race prediction performance was independent of whether the density model had been exposed to race variation during training or not. The race prediction results are summarised in Table 7.

When inspecting the t-SNE plots for the three density prediction models, we observed a clear alignment of the feature representations with the BI-RADS density classes (cf. Fig. 3a–f). For all three models, there was a gradual transition from non-dense to extremely dense samples. When overlaying race information, no obvious associations were found between race and density, independent of whether the underlying density model had been trained on a single or multiple racial groups. Samples from different races are distributed across the density feature space, with no evident

**Table 6 | Generalisation of density prediction performance across races**

| Model (Training set) | 4-Class accuracy (95% CI) | | |
|---|---|---|---|
| | Black | White | Asian |
| Model A (Black-only) | 81.6% (80.4–82.6) | 82.5% (81.4–83.6) | 82.6% (80.3–84.6) |
| Model B (White-only) | 80.2% (78.7–81.6) | 82.0% (80.5–83.1) | 81.7% (79.7–83.6) |
| Model C (Black/White) | 81.9% (80.8–83.3) | 83.0% (82.1–84.5) | 82.2% (80.4–84.0) |

Model A is trained on data from Black participants only; Model B is trained on data from White participants only; Model C is trained on data from both Black and White participants.

grouping. The t-SNE plots for the model trained from scratch for race prediction demonstrated a clear separation of data samples from Black and White participants, highlighting that race information was generally encoded in the input images. When overlaying breast density, however, no obvious associations between race and breast density were observed (cf. Fig. 3g, h).

## Discussion

In this study, we aimed to develop and evaluate a deep learning-based BI-RADS density prediction model that generalises across imaging techniques and, importantly, across race. Our primary model trained on both FFDM and 2DS images and data from Black and White participants achieved state-of-the-art performance compared to prior work[14–16,18]. While a direct comparison to previously reported results is indicative only, due to the use of different datasets, we are confident that our results on large scale, racially and ethnically diverse test sets are representative of real-world populations and would translate to clinical settings. Previous work found 2DS images to provide poorer quality for assessing parenchymal tissue in comparison to standard FFDM[21]. Nonetheless, our models showed robust performance across both imaging types. We believe this is due to the

**Table 7 | Race prediction performance across different models**

| Model | Accuracy (95% CI) | AUROC (95% CI) |
| --- | --- | --- |
| Race prediction model | 86.7% (86.0–87.4) | 95.1% (94.7–95.5) |
| Fine-tuned Model A | 74.2% (73.3–75.3) | 80.8% (80.0–81.9) |
| Fine-tuned Model B | 75.8% (74.8–76.8) | 82.8% (81.9–83.6) |
| Fine-tuned Model C | 73.2% (72.2–74.3) | 79.2% (78.4–80.3) |

First row shows the performance of a model trained specifically for race prediction. Rows two to four show the performance for race prediction of models that were built by fine-tuning the model backbones of breast density prediction models. Density models A and B were originally trained for density prediction using data from a single race group only.

inherent generalisability of the model being trained on large scale, diverse, and heterogeneous data allowing it to pick up consistent anatomical patterns present in both imaging techniques. It is also important to note that this is the first deep learning-based model that generalises across FFDM and 2DS without any modifications. This implies that a single model could be deployed, independent of the imaging technique used at a specific clinical site. Previous work required additional fine-tuning to achieve a similar level of generalisation which would result in different models being used for different imaging techniques[17]. Interesting to note that we observed an overall improvement in performance on FFDM when including 2DS training data. This suggests that there is a mutual benefit of training jointly on both imaging techniques. The robust performance of our model could address the substantial inter-reader variability observed in human readers[7,8].

A key strength of our study is the comprehensive performance and bias analysis. Firstly, our quantitative analysis of breast density prediction using different models showed consistent performance across race, including subgroups that were never seen during model development. Our experiments with models trained on different subsets of the population suggest that the density prediction performance is unbiased and equitable across groups. This is important for the trustworthy and ethical use of such AI technology, and an important finding in the context of recent concerns that AI may amplify health disparities[37–39]. Previous studies have shown disparities in the provision of breast cancer care related to race, ethnicity, and other demographic characteristics[40–43]. Here, an automated, objective method for breast density prediction could alleviate some of these disparities, for example, when assessing personalised cancer risk.

In our bias investigation, we confirmed that race information is strongly encoded in the input images by demonstrating that a model specifically trained for race prediction obtains very high accuracy. The ability to predict race from the mammographic images poses a risk as such information could be exploited in downstream model development[27]. Previous studies observed spurious and non-spurious correlations between breast density and race in data collections[22–26]. Asian participants tend to have more dense breasts than either Black or White participants[22], which we also

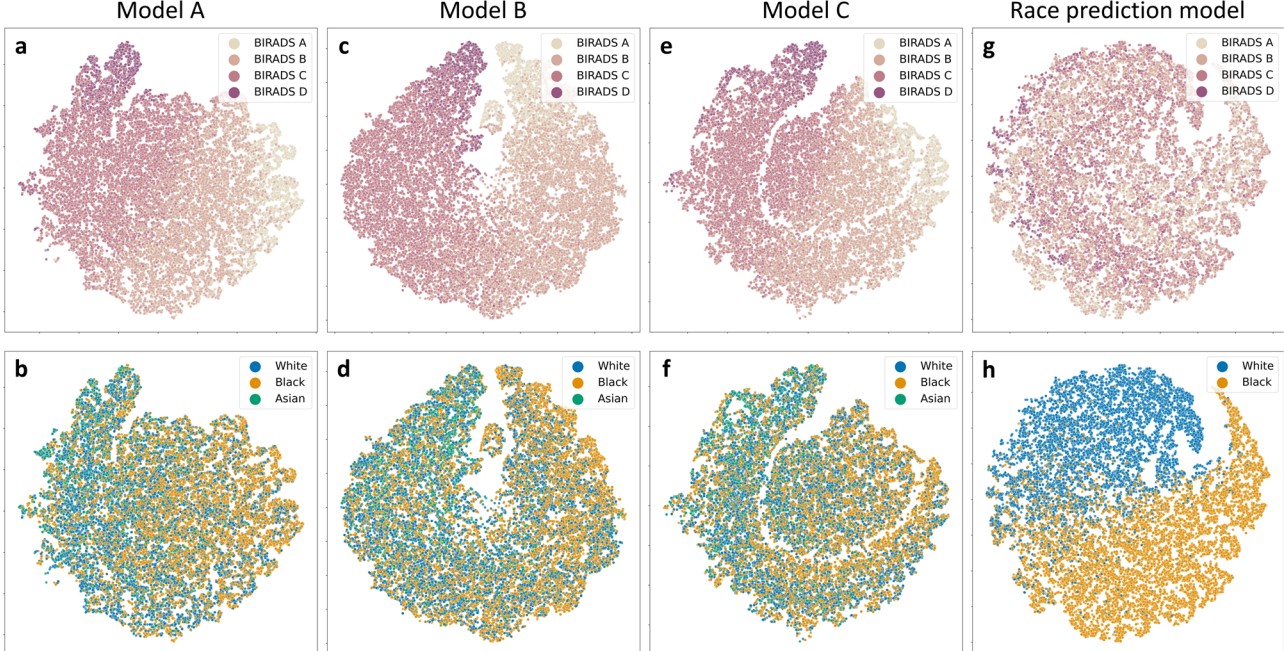

**Fig. 3 | t-SNE plots for density and race prediction models. a–f** Feature embeddings for different density prediction models. **a, b** Model A was trained on data from Black participants; **c, d** Model B was trained on data from White participants; **e, f** Model C was trained on data from both Black and White participants. **a, c, e** t-SNE plots with BI-RADS information overlaid. **b, d, f** t-SNE plots with race information overlaid. The density models encode a gradual transition across BI-RADS density classes while no obvious relationships appear between density and race information. **g, h** Feature embeddings for the race prediction model. **g** t-SNE plots with BI-RADS information overlaid. **h** t-SNE plots with race information overlaid. Here, race is well separated in the model's feature space, but no obvious associations appear between race and breast density.

observed in our study data (cf. Table 3). Given the ability of deep learning models to recognise race, a density prediction model may pick up such correlations from the historic training data. To investigate this, we conducted a detailed analysis of race prediction from fine-tuned density models where the results suggest that there is no difference in performance between a density backbone that is trained on a single race compared to a density backbone that was exposed to race variation during training. This may suggest that while some imaging information that can be leveraged for race prediction is present in the density feature representations, this exists even for models trained on a single race. Our model inspection on the feature embeddings did not detect direct associations between density and race. Together with the equitable prediction performance across races observed for differently trained density models, these findings provide confidence that racial information is not utilised for the density prediction task[31].

Our work has a number of limitations. First, while the imaging studies were collected on different devices these were from a single hardware vendor. Demonstrating generalisability across multiple hardware vendors, and also to full DBT scans, will be the focus of future work. Second, while our cohort was large and racially and ethnically diverse, all of our participants' data were from one clinical institute operating across several hospitals but within a single national breast cancer screening programme. Demonstrating generalisability on a large scale, international cohort, where geographic characteristics and specifics in the local healthcare provision may contribute additional variability, would complement the evidence presented in this work. We should note that for the purpose of this study it was important to isolate the effects of imaging technique and participant's race, and therefore the use of an otherwise homogeneous dataset where the factors of variation were limited was beneficial. Lastly, we should highlight the inherent limitations when studying the role of race. We used three self-reported racial groupings based on the data available in electronic health records. These groups may exhibit some level of inconsistency due to racial identity being influenced by an interaction between social, political, and legal constructs. Furthermore, future work should attempt to include other under-represented groups, and collect detailed information about other factors that may impact subgroup performance[44]. We believe that a detailed bias analysis is instrumental and mandatory to gain the trust of the relevant stakeholders for the use of automated computational tools in the provision of healthcare.

## Data availability
This study made use of a data sample from the EMory BrEast imaging Dataset (EMBED). Access to this data is provided upon request. Contact email: hari.trivedi@emory.edu. More details can be found on https://registry.opendata.aws/emory-breast-imaging-dataset-embed/. Source data for generating Figs. 2 and 3 are provided as Supplementary Data. Additional information related to this study is available on request to the corresponding author.

## Code availability
The code used to process the raw data and to develop the deep learning models is integrated within a commercial production system and therefore cannot be released in full. The implementation details provided in the Methods section are sufficient to replicate the deep learning models with open-source frameworks such as TensorFlow or PyTorch. We provide an example implementation published on Zenodo[45] which demonstrates the training and testing of state-of-the-art convolutional neural networks which build the core component of most commercially available breast imaging AI systems.

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

## Author contributions

G.K., H.T., B.G., T.R., and P.K. conceptualised and designed the study. G.K. implemented the deep learning models and conducted the experiments. G.K. and B.G. performed the statistical analysis. G.K., B.G., and H.T. interpreted the results and verified the underlying data. H.T. collected the data. G.K., H.T., and R.P. performed data curation and pre-processing. H.T. and M.S.N. provided medical expertise and contributed to the clinical literature review. T.R. and P.K. provided administrative support and project supervision. G.K. and B.G. wrote the initial manuscript. All authors edited and reviewed the manuscript and approved the final version.

## Competing interests

G.K., R.P., T.R., P.K., and B.G. are employees of Kheiron Medical Technologies with stock options as part of the standard compensation package. H.T. and M.S.N. declare no competing interests.
