## [Peer Review File · Communications Medicine]

Reviewers' comments:

Reviewer #1 (Remarks to the Author):

This work aims to study the generalisability of deep learning-based breast density prediction across imaging techniques [full-field digital mammography (FFDM) and two-dimensional synthetic (2DS) mammography] and across races. Experiments show that the proposed DL system trained on FFDM-only generalizes well on unseen 2DS data. Racial subgroup analysis showed unbiased performance across Black, White, and Asian participants. This is the first work that aims to investigate the potential effect of race and multiple imaging modalities on the generalisability of the deep learning system for breast density prediction.

The paper is well-written and shows interesting findings. My minor comments are below:

1. The authors claimed that the primary model trained on both FFDM and 2DS images and data from Black and White participants achieved state-of-the-art performance compared to prior work [References 14, 15, 16, 17, 18]. However, this work aimed to study the generalisability of deep learning-based breast density prediction across imaging techniques races, not focusing on optimizing deep learning systems to reach or surpass SOTA approaches. I am wondering why a simple design such as ResNet-34 network architecture with a typical training methodology can surpass SOTA approaches.
2. The proposed models showed robust performance across both imaging types, in particular, generalizes across FFDM and 2DS without any modifications, meanwhile a previous work [Reference 21] found 2DS images to provide poorer quality for assessing parenchymal tissue in comparison to standard FFDM. There are any major differences between the two experimental settings?
3. The key factor that leads to domain shift problems in medical imaging research is the imaging data were collected from multiple scanners with different hardware vendors, while the imaging studies in this study were collected on different devices these were from a single hardware vendor. This may be the reason that makes the proposed models showed robust performance across both imaging types.

Reviewer #2 (Remarks to the Author):

The study is interesting in using deep learning for the breast density assessment in different imaging techniques (FFDM and 2DS) and different races. The large number of samples is an advantage of this study. Nevertheless, certain details remain unclear, and it would be greatly beneficial if the authors could provide additional information:

- Table 1: Number of participants in Black, White and Asian were 11663, 9824, and 1570, but the total of BIRADS density cases for each race in table 3 seems to be not matching with the numbers in table 1.

- The data analysis does not report the P value so wonder if the accuracy of deep learning models were or were not significantly different among races and imaging techniques?

Response to reviewers

Many thanks for taking the time to assess our manuscript. We appreciate the positive feedback and valuable comments. We have incorporated the suggestions into the revised version of our manuscript. Please find our point-by-point responses below. In the revised version, all changes are highlighted in blue.

Reviewer 1

Comment 1: This work aims to study the generalisability of deep learning-based breast density prediction across imaging techniques [full-field digital mammography (FFDM) and two-dimensional synthetic (2DS) mammography] and across races. Experiments show that the proposed DL system trained on FFDM-only generalizes well on unseen 2DS data. Racial subgroup analysis showed unbiased performance across Black, White, and Asian participants. This is the first work that aims to investigate the potential effect of race and multiple imaging modalities on the generalisability of the deep learning system for breast density prediction.

The paper is well-written and shows interesting findings.

Response: We thank the reviewer for the positive remarks.

Comment 2: The authors claimed that the primary model trained on both FFDM and 2DS images and data from Black and White participants achieved state-of-the-art performance compared to prior work [References 14, 15, 16, 17, 18]. However, this work aimed to study the generalisability of deep learning-based breast density prediction across imaging techniques races, not focusing on optimizing deep learning systems to reach or surpass SOTA approaches. I am wondering why a simple design such as ResNet-34 network architecture with a typical training methodology can surpass SOTA approaches.

Response: The current literature detailing SOTA performance for breast density prediction on both FFDM and 2DS studies is Matthews et al. [17]. Their methodology also utilised a ResNet-34 network architecture, with a standard deep learning training methodology. We believe that our approach is comparable, and on some metrics SOTA, due to utilisation of a simple ensemble approach known as Stochastic-Weight-Averaging (described in the Implementation Details section). In the revised manuscript, we added a note in the section 'Implementation details' highlighting that a similar deep learning model was used by Matthews et al. as follows:

All deep learning models in this work were based on the ResNet-34 architecture,³⁰ implemented in TensorFlow 2. The same architecture was used in a recent study demonstrating state-of-the-art performance for breast density prediction.¹⁷

Comment 3: The proposed models showed robust performance across both imaging types, in particular, generalizes across FFDM and 2DS without any modifications, meanwhile a previous work [Reference 21] found 2DS images to provide poorer quality for assessing

parenchymal tissue in comparison to standard FFDM. There are any major differences between the two experimental settings?

Response: Thanks for pointing this out. The work of Nelson et al. [21], focused on characterising the differences between FFDM and C-View images on breast phantoms based on standard image analysis techniques, such as resolution, contrast, and signal-to-noise ratio. While our work found some minor degradation in performance when applying an FFDM trained model to C-View images, we believe the overall robust performance is due to the inherent generalisability of the DL model to pick up on consistent anatomical patterns present in both, facilitated by training on large scale, diverse, and heterogeneous data. We have added a remark to the Discussion section as follows:

Previous work found 2DS images to provide poorer quality for assessing parenchymal tissue in comparison to standard FFDM.²¹ Nonetheless, our models showed robust performance across both imaging types. We believe this is due to the inherent generalisability of the model being trained on large scale, diverse, and heterogeneous data allowing it to pick up consistent anatomical patterns present in both imaging techniques.

Comment 4: The key factor that leads to domain shift problems in medical imaging research is the imaging data were collected from multiple scanners with different hardware vendors, while the imaging studies in this study were collected on different devices these were from a single hardware vendor. This may be the reason that makes the proposed models showed robust performance across both imaging types.

Response: This is an excellent point which we also highlight in the Discussion section as a limitation of our study as follows:

First, while the imaging studies were collected on different devices these were from a single hardware vendor. Demonstrating generalisability across multiple hardware vendors, and also to full DBT scans, will be the focus of future work.

It is important to note, however, that the use of data from a single vendor allowed us to specifically isolate the effects of race and imaging techniques (FFDM vs 2DS), which is the primary focus of our work and has not been studied elsewhere. These effects would be difficult to disentangle in multi-vendor datasets.

Reviewer 2

Comment 1: The study is interesting in using deep learning for the breast density assessment in different imaging techniques (FFDM and 2DS) and different races. The large number of samples is an advantage of this study.

Response: Many thanks for the positive feedback.

Comment 2: *Table 1:* Number of participants in Black, White and Asian were 11663, 9824, and 1570, but the total of BIRADS density cases for each race in *table 3* seems to be not matching with the numbers in table 1.

Response: The participants row in table 1 refers to the number of participants present in the entire study cohort. The counts per BIRADS density class presented in table 3 are on a study level, and only refer to the test set (on which we determine the model performance). Please see table 2 for a breakdown of the training, validation, and test sets. The counts per density class in table 3 sum up to the total number of studies in the test set (reported in the last column of table 2). To avoid confusion, we have now clarified in the main text that table 3 refers to the test set.

Comment 3: The data analysis does not report the P value so I wonder if the accuracy of deep learning models were or were not significantly different among races and imaging techniques?

Response: Thanks for pointing this out. We have now added an additional set of analyses to measure the equivalence of groups more rigorously. We focus on two sets of comparisons:

1. On a model basis, evaluate how similarly each racial subgroup performs w.r.t to one another.
2. On a race basis, evaluate how similarly each model performs w.r.t to one another.

For all configurations, we employed a ‘two-one-sided t-tests’ (TOST) procedure to test equivalence, with an absolute equivalence margin of 2% for the difference in the average 4-class accuracy of compared groups. In this setup, our null hypothesis, H_0 , assumes that the difference in our mean accuracy between populations is greater than our 2% bound. In equivalence testing, a p-value less than 0.05 (at a confidence level of 95%) allows us to reject the null hypothesis, and thus implies that the performance across both populations is within this equivalence margin. The single p-value reported in the tables below was the largest value from either side of the TOST. All equivalence p-values were calculated from 1,000 bootstrapped samples, with each bootstrapped sample consisting of 5,000 studies if from Black or White participants, and 500 studies if from Asian participants, all sampled with replacement.

The following tables have been added to the Supplementary Information and the description of the statistical analysis in the main manuscript has been revised accordingly.

(1) Per model, across races p-value tables;

Model A	Black	White	Asian
Black	< 10 ⁻⁵	< 10 ⁻⁵	< 10 ⁻⁵
White	< 10 ⁻⁵	< 10 ⁻⁵	< 10 ⁻⁵
Asian	< 10 ⁻⁵	< 10 ⁻⁵	< 10 ⁻⁵

P-value table : Model A is trained on data from Black participants only. TOST p-value less than 0.05 for all sub-group comparisons at an equivalence margin of 2%.

Model B	Black	White	Asian
Black	< 10 ⁻⁵	0.002	< 10 ⁻⁵
White	9 x 10 ⁻⁴	< 10 ⁻⁵	< 10 ⁻⁵
Asian	< 10 ⁻⁵	< 10 ⁻⁵	< 10 ⁻⁵

P-value table : Model B is trained on data from White participants only. TOST p-value less than 0.05 for all sub-group comparisons at an equivalence margin of 2%.

Model C	Black	White	Asian
Black	< 10 ⁻⁵	< 10 ⁻⁵	< 10 ⁻⁵
White	< 10 ⁻⁵	< 10 ⁻⁵	< 10 ⁻⁵
Asian	< 10 ⁻⁵	< 10 ⁻⁵	< 10 ⁻⁵

P-value table : Model C is trained on data from both White and Black participants. TOST p-value less than 0.05 for all sub-group comparisons at an equivalence margin of 2%.

(2) Per race, across models;

Black	Model A	Model B	Model C
Model A	< 10 ⁻⁵	< 10 ⁻⁵	< 10 ⁻⁵
Model B	< 10 ⁻⁵	< 10 ⁻⁵	< 10 ⁻⁵
Model C	< 10 ⁻⁵	< 10 ⁻⁵	< 10 ⁻⁵

P-value table : Model A is trained on data from Black participants only, model B is trained on data from White participants only, and model C is trained on data from both White and Black participants. TOST p-value less than 0.05 for all sub-group comparisons at an equivalence margin of 2%.

White	Model A	Model B	Model C
Model A	< 10 ⁻⁵	< 10 ⁻⁵	< 10 ⁻⁵
Model B	< 10 ⁻⁵	< 10 ⁻⁵	< 10 ⁻⁵
Model C	< 10 ⁻⁵	< 10 ⁻⁵	< 10 ⁻⁵

P-value table : Model A is trained on data from Black participants only, model B is trained on data from White participants only, and model C is trained on data from both White and Black participants. TOST p-value less than 0.05 for all sub-group comparisons at an equivalence margin of 2%.

Asian	Model A	Model B	Model C
Model A	< 10 ⁻⁵	2 x 10 ⁻⁵	< 10 ⁻⁵
Model B	3 x 10 ⁻⁴	< 10 ⁻⁵	5 x 10 ⁻⁴
Model C	< 10 ⁻⁵	0.02	< 10 ⁻⁵

P-value table : Model A is trained on data from Black participants only, model B is trained on data from White participants only, and model C is trained on data from both White and Black participants. TOST p-value less than 0.05 for all sub-group comparisons at an equivalence margin of 2%.

REVIEWERS' COMMENTS:

Reviewer #1 (Remarks to the Author):

Thank you for considering and addressing all of my comments and suggestions. I have no more comments and I support the publication of this work.

Thanks!

Response to reviewers

Many thanks for taking the time to assess our manuscript. We appreciate the positive feedback and valuable comments. We have incorporated the suggestions into the revised version of our manuscript. Please find our point-by-point responses below. In the revised version, all changes are highlighted in blue.

Reviewer 1

Comment 1: This work aims to study the generalisability of deep learning-based breast density prediction across imaging techniques [full-field digital mammography (FFDM) and two-dimensional synthetic (2DS) mammography] and across races. Experiments show that the proposed DL system trained on FFDM-only generalizes well on unseen 2DS data. Racial subgroup analysis showed unbiased performance across Black, White, and Asian participants. This is the first work that aims to investigate the potential effect of race and multiple imaging modalities on the generalisability of the deep learning system for breast density prediction.

The paper is well-written and shows interesting findings.

Response: We thank the reviewer for the positive remarks.

Comment 2: The authors claimed that the primary model trained on both FFDM and 2DS images and data from Black and White participants achieved state-of-the-art performance compared to prior work [References 14, 15, 16, 17, 18]. However, this work aimed to study the generalisability of deep learning-based breast density prediction across imaging techniques races, not focusing on optimizing deep learning systems to reach or surpass SOTA approaches. I am wondering why a simple design such as ResNet-34 network architecture with a typical training methodology can surpass SOTA approaches.

Response: The current literature detailing SOTA performance for breast density prediction on both FFDM and 2DS studies is Matthews et al. [17]. Their methodology also utilised a ResNet-34 network architecture, with a standard deep learning training methodology. We believe that our approach is comparable, and on some metrics SOTA, due to utilisation of a simple ensemble approach known as Stochastic-Weight-Averaging (described in the Implementation Details section). In the revised manuscript, we added a note in the section 'Implementation details' highlighting that a similar deep learning model was used by Matthews et al. as follows:

All deep learning models in this work were based on the ResNet-34 architecture,³⁰ implemented in TensorFlow 2. The same architecture was used in a recent study demonstrating state-of-the-art performance for breast density prediction.¹⁷

Comment 3: The proposed models showed robust performance across both imaging types, in particular, generalizes across FFDM and 2DS without any modifications, meanwhile a previous work [Reference 21] found 2DS images to provide poorer quality for assessing

parenchymal tissue in comparison to standard FFDM. There are any major differences between the two experimental settings?

Response: Thanks for pointing this out. The work of Nelson et al. [21], focused on characterising the differences between FFDM and C-View images on breast phantoms based on standard image analysis techniques, such as resolution, contrast, and signal-to-noise ratio. While our work found some minor degradation in performance when applying an FFDM trained model to C-View images, we believe the overall robust performance is due to the inherent generalisability of the DL model to pick up on consistent anatomical patterns present in both, facilitated by training on large scale, diverse, and heterogeneous data. We have added a remark to the Discussion section as follows:

Previous work found 2DS images to provide poorer quality for assessing parenchymal tissue in comparison to standard FFDM.²¹ Nonetheless, our models showed robust performance across both imaging types. We believe this is due to the inherent generalisability of the model being trained on large scale, diverse, and heterogeneous data allowing it to pick up consistent anatomical patterns present in both imaging techniques.

Comment 4: The key factor that leads to domain shift problems in medical imaging research is the imaging data were collected from multiple scanners with different hardware vendors, while the imaging studies in this study were collected on different devices these were from a single hardware vendor. This may be the reason that makes the proposed models showed robust performance across both imaging types.

Response: This is an excellent point which we also highlight in the Discussion section as a limitation of our study as follows:

First, while the imaging studies were collected on different devices these were from a single hardware vendor. Demonstrating generalisability across multiple hardware vendors, and also to full DBT scans, will be the focus of future work.

It is important to note, however, that the use of data from a single vendor allowed us to specifically isolate the effects of race and imaging techniques (FFDM vs 2DS), which is the primary focus of our work and has not been studied elsewhere. These effects would be difficult to disentangle in multi-vendor datasets.

Reviewer 2

Comment 1: The study is interesting in using deep learning for the breast density assessment in different imaging techniques (FFDM and 2DS) and different races. The large number of samples is an advantage of this study.

Response: Many thanks for the positive feedback.

Comment 2: *Table 1:* Number of participants in Black, White and Asian were 11663, 9824, and 1570, but the total of BIRADS density cases for each race in *table 3* seems to be not matching with the numbers in *table 1*.

Response: The participants row in *table 1* refers to the number of participants present in the entire study cohort. The counts per BIRADS density class presented in *table 3* are on a study level, and only refer to the test set (on which we determine the model performance). Please see *table 2* for a breakdown of the training, validation, and test sets. The counts per density class in *table 3* sum up to the total number of studies in the test set (reported in the last column of *table 2*). To avoid confusion, we have now clarified in the main text that *table 3* refers to the test set.

Comment 3: The data analysis does not report the P value so I wonder if the accuracy of deep learning models were or were not significantly different among races and imaging techniques?

Response: Thanks for pointing this out. We have now added an additional set of analyses to measure the equivalence of groups more rigorously. We focus on two sets of comparisons:

1. On a model basis, evaluate how similarly each racial subgroup performs w.r.t to one another.
2. On a race basis, evaluate how similarly each model performs w.r.t to one another.

For all configurations, we employed a ‘two-one-sided t-tests’ (TOST) procedure to test equivalence, with an absolute equivalence margin of 2% for the difference in the average 4-class accuracy of compared groups. In this setup, our null hypothesis, H_0 , assumes that the difference in our mean accuracy between populations is greater than our 2% bound. In equivalence testing, a p-value less than 0.05 (at a confidence level of 95%) allows us to reject the null hypothesis, and thus implies that the performance across both populations is within this equivalence margin. The single p-value reported in the tables below was the largest value from either side of the TOST. All equivalence p-values were calculated from 1,000 bootstrapped samples, with each bootstrapped sample consisting of 5,000 studies if from Black or White participants, and 500 studies if from Asian participants, all sampled with replacement.

The following tables have been added to the Supplementary Information and the description of the statistical analysis in the main manuscript has been revised accordingly.

(1) Per model, across races p-value tables;

Model A	Black	White	Asian
Black	$< 10^{-5}$	$< 10^{-5}$	$< 10^{-5}$
White	$< 10^{-5}$	$< 10^{-5}$	$< 10^{-5}$
Asian	$< 10^{-5}$	$< 10^{-5}$	$< 10^{-5}$

P-value table : Model A is trained on data from Black participants only. TOST p-value less than 0.05 for all sub-group comparisons at an equivalence margin of 2%.

Model B	Black	White	Asian
Black	$< 10^{-5}$	0.002	$< 10^{-5}$
White	9×10^{-4}	$< 10^{-5}$	$< 10^{-5}$
Asian	$< 10^{-5}$	$< 10^{-5}$	$< 10^{-5}$

P-value table : Model B is trained on data from White participants only. TOST p-value less than 0.05 for all sub-group comparisons at an equivalence margin of 2%.

Model C	Black	White	Asian
Black	$< 10^{-5}$	$< 10^{-5}$	$< 10^{-5}$
White	$< 10^{-5}$	$< 10^{-5}$	$< 10^{-5}$
Asian	$< 10^{-5}$	$< 10^{-5}$	$< 10^{-5}$

P-value table : Model C is trained on data from both White and Black participants. TOST p-value less than 0.05 for all sub-group comparisons at an equivalence margin of 2%.

(2) Per race, across models;

Black	Model A	Model B	Model C
Model A	$< 10^{-5}$	$< 10^{-5}$	$< 10^{-5}$
Model B	$< 10^{-5}$	$< 10^{-5}$	$< 10^{-5}$
Model C	$< 10^{-5}$	$< 10^{-5}$	$< 10^{-5}$

P-value table : Model A is trained on data from Black participants only, model B is trained on data from White participants only, and model C is trained on data from both White and Black participants. TOST p-value less than 0.05 for all sub-group comparisons at an equivalence margin of 2%.

White	Model A	Model B	Model C
Model A	$< 10^{-5}$	$< 10^{-5}$	$< 10^{-5}$
Model B	$< 10^{-5}$	$< 10^{-5}$	$< 10^{-5}$
Model C	$< 10^{-5}$	$< 10^{-5}$	$< 10^{-5}$

P-value table : Model A is trained on data from Black participants only, model B is trained on data from White participants only, and model C is trained on data from both White and Black participants. TOST p-value less than 0.05 for all sub-group comparisons at an equivalence margin of 2%.

Asian	Model A	Model B	Model C
Model A	$< 10^{-5}$	2×10^{-5}	$< 10^{-5}$
Model B	3×10^{-4}	$< 10^{-5}$	5×10^{-4}
Model C	$< 10^{-5}$	0.02	$< 10^{-5}$

P-value table : Model A is trained on data from Black participants only, model B is trained on data from White participants only, and model C is trained on data from both White and Black participants. TOST p-value less than 0.05 for all sub-group comparisons at an equivalence margin of 2%.